Identification of KIF4A as a pan-cancer diagnostic and prognostic biomarker via bioinformatics analysis and validation in osteosarcoma cell lines

Pan Jiankang 1
Lei Xiaohua leixiaohua2011@sina.com 2
Mao Xinzhan xinzhan.mao@csu.edu.cn 1
1 Department of Orthopedics, the Second Xiangya Hospital, Central South University , Changsha , Hunan , China
2 Department of Hepato-Biliary-Pancreatic Surgery, the First Affiliated Hospital of University of South China , Hengyang , Hunan , China
Soares Paula
Electronic publication date: 2021 May 21
Publication date: 2021
Volume: 9
Electronic Location ID: e11455
Received 2020 Dec 3; Accepted 2021 Apr 23
Copyright: ©2021 Pan et al.
Copyright year: 2021
Copyright holder: Pan et al.
License: This is an open access article distributed under the terms of the Creative Commons Attribution License, which permits unrestricted use, distribution, reproduction and adaptation in any medium and for any purpose provided that it is properly attributed. For attribution, the original author(s), title, publication source (PeerJ) and either DOI or URL of the article must be cited.
License URL: https://creativecommons.org/licenses/by/4.0/

Keywords: Bioinformatics analysis, Chromosome instability biomarker, KIF4A, OS

Funding: Hunan Provincial Health and Family Planning Commission Research Project 20180343 Health Commission of Hunan Province Research Project 20200284 This work is supported by the following grants: Hunan Provincial Health and Family Planning Commission Research Project, Grant No. 20180343. Health Commission of Hunan Province Research Project Grant No. 20200284. The funders had no role in study design, data collection and analysis, decision to publish, or preparation of the manuscript.

==============================
Background

Cancer is a disease of abnormal cell proliferation caused by abnormal expression of cancer-related genes. However, it is still difficult to distinguish benign and malignant lesions in many cases. KIF4A has been reported to be associated with a variety of cancer lesions. We aimed to explore whether KIF4A could be used as a biomarker of pan-cancer diagnostic.

Methods

We identified twenty-eight cell cycle-related genes that were overexpressed in no less than ten types of cancer. We determined KIF4A mRNA and protein expression in osteosarcoma (OS) cells. Furthermore, to determine the effect of KIF4A in OS, we silenced KIF4A in OS cells and detected cell viability, colony formation, invasion, migration, apoptosis and cell cycle parameters.

Results

KIF4A exhibited upregulated expression in eleven types of cancer. Cell cycle-related genes are extensively overexpressed in various types of cancers. KIF4A overexpression can serve as a diagnostic and prognostic marker in various cancers. Silencing KIF4A inhibited the viability, colony formation, invasion and migration and induced apoptosis and cell cycle arrest of OS cells. Our findings revealed that high expression of KIF4A could serve as a diagnostic and prognostic marker in OS cancers.

Conclusion

KIF4A could serve as a pan-cancer diagnostic and prognostic marker. KIF4A could be used as a novel therapeutic target for OS.

Introduction

Osteosarcoma (OS) is the most common malignant bone tumor in the world (accounting for nearly 60% of bone malignant tumors), which seriously affects the daily life of patients (Gianferante & Mirabello, 2017). Essentially, cancer is a disease of abnormal cell proliferation caused by accumulated genomic mutations (Hu et al., 2018). Previous studies in molecular medicine have divided cancers into increasingly specific molecular subtypes according to their specific genomic status or transcriptome pattern, which has greatly improved the prognosis of many cancer types (Yang, Wang & Zheng, 2019; Hu et al., 2019; Knight et al., 2020). However, it is still difficult to distinguish benign and malignant lesions in many cases. There is an urgent need for effective treatment. Therefore, it is necessary to identify different cancers according to the current trend of developing more precise drugs.

Transcriptome analysis to identify differential gene expression between tumor and paired normal tissues can help to reveal the key process of cell proliferation in cancer. RNA-seq data from the Cancer Genome Atlas (TCGA) dataset could systematically analyze the expression profiles of bidirectional genes and gene pairs in cancer (Tu & Li, 2019). We plan to screen RNA-seq data covering dozens of cancers and matched normal tissues from TCGA. We will further use bioinformatics to analyze whether KIF4A is involved in chromosome segregation and whether it is abnormally expressed in various types of cancer.

The KIF4A gene encodes the chromokinesin protein KIF4A, an Adenosine triphosphate (ATP) dependent molecular motor that promotes mitotic chromosome condensation and segregation (Mazumdar, Sundareshan & Misteli, 2004). KIF4A can also directly bind to chromatin and participate in DNA damage repair by associating with BRCA2 (Wu & Chen, 2008; Wu et al., 2008). Many studies have noted the presence of upregulated expression of KIF4A in various cancer tissues and its positive correlation with poor prognosis (Hou et al., 2017; Narayan et al., 2007; Taniwaki et al., 2007; Minakawa et al., 2013).

However, the role of KIF4A in OS remains unknown. We studied the role of KIF4A in OS cells. Our results indicated that KIF4A could be a predictive factor and therapeutic target in OS.

Material and Methods

The Cancer Genome Atlas (TCGA) dataset

Data acquisition and analysis were conducted using R software (version 3.5.1 or above), unless otherwise mentioned. RNA-seq and clinical data were downloaded using the TCGA Biolinks R/Bioconductor package (version 2.10.5) (Colaprico et al., 2016; Le, 2020). Generally, we used TCGA Biolinks to download all available samples with Illumina HiSeq RNASeqV2 data from 33 cancer types.

Data analysis, gene network analysis and pathway enrichment analysis

The fragments per kilobase of transcript per million fragments mapped (FPKM) parameter is the most commonly used normalization method for analyzing RNA transcript reads. The upper quantile normalized FPKM (FPKM-UQ ) method have used the upper quantile gene count instead of the total gene count for normalization , which is considered to have superior sensitivity in the identification of gene differential expression (Hsu, 2020; Bullard, 2010). In this study, FPKM-UQ RNA-seq data were downloaded and prepared using the GDC query, GDC download, and GDC prepare functions. All analysis codes used are freely available at https://github.com/hutaobo/prognosis.

Cell culture and transfection

The normal osteoblast cell line hFOB1.19 and the human OS cell lines MG63, U2OS and HOS were purchased from the Cell Bank of the Chinese Academy of Sciences and cultured in 5% CO2 at 37 °C in DMEM (Sigma) containing 10% FBS (Gibco), 100 U/ml penicillin and 100 pg/ml streptomycin. The cells were evaluated in the logarithmic growth phase. A siRNA against KIF4A (siKIF4A) and EX-A3631-Lv105 (oe-KIF4A, GeneCopoeia, Inc.) were purchased, and the transfection was performed using Lipofectamine 2000 (Invitrogen).

Cell Counting Kit-8 (CCK-8)

To assess cell viability, cells were seeded in a 96-well plate at a density of 5 ×10 (Yang, Wang & Zheng, 2019) cells per well. At the indicated times (24, 48 and 72 h), 10 µl CCK-8 solution (Dojindo) was added to each well, and the plates were incubated at 37 °C for 2 h. The proliferation of cells was determined with a CCK-8 assay kit. The absorbance was measured at 450 nm.

Wound healing assay

The cells were seeded in 6-well plates and incubated to nearly 100% confluence. The cell monolayer was scratched with a 10 µl plastic pipette tip. The wells were washed with phosphate-buffered saline (PBS), FBS was added to the well, and the area of scratch closure was used to estimate the migratory ability with an inverted phase microscope. The percentage of wound closure was calculated as a ratio of the wound area at 24 h to that at 0 h and 48 h.

Cell colony formation

The indicated cells (200 cells/2 ml) were plated into 6-well plates and incubated for two weeks. The colonies were fixed with methanol and stained with 1% crystal violet. The absorbance was measured at 550 nm.

Flow cytometry analysis

Cell cycle parameters were detected by propidium iodide (PI) staining. Briefly, 48 h after transfection, cells were collected and fixed overnight with 90% cold ethanol at 20 °C. The next day, the cells were incubated at RT for 5 min. The cells were washed with PBS twice and then cultured with 1 ml PI staining solution (50 µg/ml; 1 mg/ml RNase A, and 0.1% Triton X-100 in PBS) in the dark for 30 min. Cell cycle proportions were detected by flow cytometry (FACS Calibur).

Cell apoptosis was detected by using Annexin V-FITC/PI double staining. Briefly, after the transfected cells were collected and washed, they were incubated with 500 µl binding buffer, 5 µl Annexin V-FITC (BD) and 5 µl propidium iodide (PI). The apoptotic rate was determined by using flow cytometry.

Western blotting

Proteins were separated using a 10% SDS-PAGE gel and then transferred to PVDF membranes. After blocking in nonfat milk, the immunoblots were incubated with primary antibodies against the following molecules: KIF4A (1:1000, ab122227, Abcam), Bax (1:2000, ab32503, Abcam), Bcl-2 (1:1000, 12789-1-AP, Proteintech), cleaved-caspase 3 (1:500, ab32042, Abcam), Wnt3a (1:1000, ab28472, Abcam), β-catenin (1:6000, 51067-2-AP, Proteintech) and p-β-catenin (1:500, PA5-36827, proteintech) were purchased. Anti-β-actin antibodies (1:5000, 66009-1-Ig, Proteintech) were used as an internal control. Protein bands were visualized using an enhanced chemiluminescence (ECL) machine (Advansta).

Transwell assay

Transwell assays were conducted to detect cell invasion. Crystalline violet was dissolved in 95% ethanol, mixed with ammonium oxalate solution, and stood for 48 h. A total of 1 × 106 cells were added to the upper chambers and cultured in 200 ml serum-free DMEM. After incubation for 48 h at 37 °C, the upper surface of the membrane was wiped with a cotton tip, and the cells on the lower membrane were fixed with 4% polyformaldehyde and stained with 0.1% crystal violet for 30 min. The absorbance was measured at 550 nm.

Real-time quantitative reverse transcription PCR (qRT-PCR) analysis

The transcription level was determined by qRT-PCR. Briefly, total RNA of the cells was dissociated with an RNA extraction kit. RNA was reverse-transcribed into cDNA with a reverse transcription kit (ComWin Biotech). Quantification of gene expression was conducted using the 2−ΔΔCT method, and the results were normalized to β-actin mRNA levels. The sequences of the primers were as follows: KIF4A: F-TGTTGGATGTGGGCCTTAGC, R-GTGACTTAGCACCCTTCTGGA; β-actin: F-ACCCTGAAGTACCCCATCGAG, R-AGCACAGCCTGGATAGCAAC.

Statistical analysis

All bioinformatic analyses were conducted using R software. The SPSS22.0 software program was used for the statistical analysis. The t-test and one-way analysis of variance (ANOVA) were used for comparisons between two groups, and comparisons among multiple groups were made using two-way ANOVA. P < 0.05 was considered statistically significant.

Results

Cell cycle-related genes are extensively overexpressed in various types of cancers

The expression of 57,035 genes in 12 types of cancer and their paired normal tissues from TCGA were analyzed. The 12 analyzed types of cancers were breast invasive carcinoma (BRCA) (The Cancer Genome Atlas Network, 2012c), kidney renal clear cell carcinoma (KIRC) (The Cancer Genome Atlas Network, 2013), lung adenocarcinoma (LUAD) (The Cancer Genome Atlas Network, 2014b), stomach adenocarcinoma (STAD) (The Cancer Genome Atlas Network, 2014a), colon adenocarcinoma (COAD) (The Cancer Genome Atlas Network, 2012b), kidney renal papillary cell carcinoma (KIRP) (Linehan et al., 2016), lung squamous cell carcinoma (LUSC) (The Cancer Genome Atlas Network, 2012a), thyroid carcinoma (THCA) (The Cancer Genome Atlas Research Network, 2014c), head and neck squamous cell carcinoma (HNSC) (The Cancer Genome Atlas Network, 2015a), liver hepatocellular carcinoma (LIHC) (The Cancer Genome Atlas Network, 2017), prostate adenocarcinoma (PRAD) (The Cancer Genome Atlas Network, 2015b), and uterine corpus endometrial carcinoma (UCEC) (Cherniack et al., 2017). The sample number for each cancer type is listed in Table 1.

Table 1 List of full names and sample numbers for each type of cancer.

Abbr.	Cancer type	N of samples	
TCGA-BRCA	Breast Invasive Carcinoma	112	
TCGA-COAD	Colon Adenocarcinoma	41	
TCGA-HNSC	Head and Neck Squamous Cell Carcinoma	43	
TCGA-KIRC	Kidney Renal Clear Cell Carcinoma	72	
TCGA-KIRP	Kidney Renal Papillary Cell Carcinoma	31	
TCGA-LIHC	Liver Hepatocellular Carcinoma	50	
TCGA-LUAD	Lung Adenocarcinoma	57	
TCGA-LUSC	Lung Squamous Cell Carcinoma	49	
TCGA-PRAD	Prostate Adenocarcinoma	52	
TCGA-STAD	Stomach Adenocarcinoma	27	
TCGA-THCA	Thyroid Carcinoma	58	
TCGA-UCEC	Uterine Corpus Endometrial Carcinoma	23	
		Total: 615	

For statistical analysis, only those genes reaching genome-wide significance were included and defined as differentially expressed genes (DEGs) (p value less than 5 × 10−8) (Panagiotou & Ioannidis, 2012). Approximately three-quarters of the genes did not show elevated expression in any cancer type. Thus, only 9% of all the analyzed genes (n = 5343) showed elevated expression in more than one type of cancer. Among them, 28 genes were found to be overexpressed in no less than 10 types of cancer (Table 2). In gastric adenocarcinoma and thyroid cancer, only 6 and 13 of 28 pan-cancer DEGs are overexpressed. In the other 10 cancer types, at least 20 DEGs are overexpressed. This difference was not caused by sample size, as thyroid carcinoma had the third largest sample size. Therefore, this finding indicates the existence of an intrinsic difference in stomach adenocarcinoma and thyroid carcinoma compared with other cancer types. The Gene Ontology (GO) molecular pathway analysis showed that the 28 pan-cancer DEGs were enriched in 116 biological processes, most of which were cell cycle-related processes (Table S1). This is no surprise, since cancer is essentially a disease of uncontrolled cell proliferation. However, only 28 DEGs of 1263 genes involved in the cell cycle have been extensively altered in different types of cancer. This suggested that these genes may be the key to cancer cell cycle regulation. The 28 selected DEGs also had strong protein–protein interactions, as plotted using STRING (Fig. 1A).

Table 2 Expression conditions of the top 28 DEGs in the 12 types of cancers investigated.

	BRCA	COAD	HNSC	KIRC	KIRP	LIHC	LUAD	LUSC	PRAD	STAD	THCA	UCEC	N	
KIF4A	+	+	+	+	+	+	+	+	+	N.S.	+	+	11	
STIL	+	+	+	+	+	+	+	+	+	+	N.S.	+	11	
TMEM132A	+	+	+	+	N.S.	N.S.	+	+	+	+	+	+	10	
TRIP13	+	+	+	+	+	+	+	+	+	+	N.S.	N.S.	10	
GTSE1	+	+	+	+	+	+	+	+	+	N.S.	N.S.	+	10	
UBE2T	+	+	+	+	+	+	+	+	N.S.	N.S.	+	+	10	
AURKA	+	+	+	+	+	+	+	+	+	N.S.	N.S.	+	10	
TPX2	+	+	+	+	+	+	+	+	+	N.S.	N.S.	+	10	
BIRC5	+	+	+	+	+	+	+	+	+	N.S.	N.S.	+	10	
ORC6	+	+	+	+	+	+	+	+	N.S.	N.S.	+	+	10	
CLSPN	+	N.S.	+	+	+	+	+	+	N.S.	+	+	+	10	
CDC45	+	+	+	+	+	+	+	+	N.S.	N.S.	+	+	10	
CDC6	+	+	+	+	+	+	+	+	N.S.	N.S.	+	+	10	
MMP11	+	+	+	+	+	+	+	+	N.S.	+	+	N.S.	10	
CDKN3	+	+	+	+	+	+	+	+	+	N.S.	N.S.	+	10	
MYBL2	+	+	+	+	+	+	+	+	+	N.S.	N.S.	+	10	
E2F1	+	+	+	+	+	+	N.S.	+	N.S.	+	+	+	10	
RNASEH2A	+	+	+	+	+	+	+	+	N.S.	N.S.	+	+	10	
ASF1B	+	+	N.S.	+	+	+	+	+	+	N.S.	+	+	10	
EZH2	+	+	N.S.	+	+	+	+	+	+	N.S.	+	+	10	
FOXM1	+	+	+	+	+	+	+	+	+	N.S.	N.S.	+	10	
CDCA3	+	+	+	+	+	+	+	+	+	N.S.	N.S.	+	10	
KIF20A	+	+	N.S.	+	+	+	+	+	+	N.S.	+	+	10	
CENPA	+	+	+	+	+	+	+	+	+	N.S.	N.S.	+	10	
KIF14	+	+	+	+	+	+	+	+	+	N.S.	N.S.	+	10	
NCAPH	+	+	+	+	+	+	+	+	+	N.S.	N.S.	+	10	
HJURP	+	+	+	+	+	+	+	+	+	N.S.	N.S.	+	10	
PKMYT1	+	+	+	+	+	+	+	+	N.S.	N.S.	+	+	10	
Notes.

+: specific gene is overexpressed in that cancer type compared to normal tissue.

N.S., not significant.

Figure 1 Result of bioinformatics analysis.

(A) Protein–protein interactions (PPI) involving the 28 differentially expressed genes (DEGs) were identified using the STRING database. The “experiment”, “database”, and “coexpression” evidence channels were chosen for network construction. Clustering was performed using the MCL algorithm with inflation parameter 10. Different colors indicate different clusters and the line thickness indicates the strength of evidence. (B–M) Expression profile of KIF4A in cancer tissues and paring normal tissues. (N–Q) Kaplan–Meier survival curves for four cancer types in regard to KIF4A expression.

KIF4A overexpression can serve as a diagnostic and prognostic marker in various cancers

Among the pan-cancer DEGs, KIF4A showed elevated expression in eleven types of cancer (Figs. 1B–1M). KIF4A was also found to be upregulated in multiple cancer types. Elevated expression of the KIF4A protein was also verified in five types of cancer using immunohistochemistry data from the Human Protein Atlas (HPA; Fig. S1).

High expression of the KIF4A gene was significantly correlated with poor prognosis in four kinds of cancer, KIRC, KIRP, LUAD, and LIHC, as shown by Kaplan–Meier survival analysis (Figs. 1N–1Q).

Expression of KIF4A in OS cell lines

To further validate the bioinformatics analysis results, the expression of KIF4A in OS cell lines was detected. The above analysis results found that KIF4A is highly expressed in osteosarcoma. We determined the mRNA and protein expression of KIF4A in three OS cell lines (MG63, U2OS and HOS) and normal osteoblast cells using RT-PCR and Western blotting (Fig. 2). The results showed that the level of mRNA and protein of KIF4A in osteosarcoma cell lines were significantly higher than those of normal osteoblasts hFOB1.19. The results of TCGA data analysis were consistent with the results of cell experiment.

Figure 2 KIF4A mRNA and protein expression were down-regulated in MG63 and U2OS cells transfected with si-KIF4A.

(A) KIF4A mRNA level was higher in OS cells than in the normal osteoblast hFOB1.19. (B) KIF4A protein expression was significantly higher in OS cell lines than in the normal osteoblast hFOB1.19. (C) PCR assay confirmed that KIF4A mRNA expression were downregulated in MG63 and U2OS cells transfected with si-KIF4A. * p < 0.05, *** p < 0.001.

Effect of silencing KIF4A on the viability, colony formation, invasion and migration of OS cells

Among the three OS cell lines, KIF4A expression was highest in MG63 and U2OS cells; therefore, these two lines were chosen for subsequent assays. The silencing plasmid and overexpression plasmid was transfected into MG63 and U20S cells respectivel. The results showed that the transfection was successful (Figs. 3A–3D). The CCK-8 assay revealed that decreased KIF4A expression markedly suppressed the viability of MG63 and U2OS cells. Overexpression of KIF4A enhanced cell viability (Fig. 3E and 3F). Colony formation experiments showed that after knocking out KIF4A, the cloning ability of MG63 and U2OS cells was greatly reduced. After overexpression of KIF4A, the clonality of cells was promoted (Figs. 3G–3I), which indicated that KIF4A increased the stemness of OS cells. Transwell assays revealed that the silencing of KIF4A strikingly decreased the invasion ability of MG63 and U2OS cells (Fig. 4A and 4B). However, the cell invasion ability of the oe-KIF4A group was significantly increased. The date of wound healing assay showed that the migration ability of MG63 and U2OS cells was significantly decreased after KIF4A knockdown. After KIF4A was up-regulated, the cell migration ability increased significantly (Figs. 4C–4F). In short, KIF4A could affect the activity of MG63 and U2OS cells.

Figure 3 KIF4A could affect OS cell viability and colony formation and migration.

(A–D) Expression of KIF4A in cells was detected by RT-PCR and Western blot. (E and F) CCK8 was used to detect cell proliferation. (G–I) Colony formation results confirmed that in MG63 and U2OS cell lines, OS cell colony formation were inhibited by silencing KIF4A. *P < 0.05 compared with Control, #P < 0.05 compared with si-NC, &P < 0.05 compared with oe-NC.

Figure 4 Silencing KIF4A inhibited migration of Osteosarcoma cells.

(A and B) Transwell assay was used to test the invasion. Magnification ×100, scale bar = 100 µm. (C–F) Wound healing assay was utilized to detect the migration. Magnification ×100, scale bar = 100 µm. *P < 0.05 compared with Control, #P < 0.05 compared with si-NC, &P < 0.05 compared with oe-NC.

Effect of KIF4A silencing on the apoptosis and cell cycle of OS cells

The above results indicate that KIF4A could affect the activity of MG63 and U20S cells. We further investigated whether apoptosis would be affected. Flow cytometry results showed that KIF4A silencing could induce apoptosis, leading to cell cycle arrest in G1 phase. KIF4A overexpression can inhibit cell apoptosis and reduce the proportion of cells in G1 phase (Figs. 5A–5X). The expression of apoptosis-related proteins (caspase-3, Bax and Bcl-2) was determined by Western blot analysis. The Western blot results showed that silencing of KIF4A significantly upregulated the protein levels of Bax and caspase-3 and decreased the Wnt3a, p-β-catenin and Bcl-2 ratio. The data overexpressing KIF4A showed the opposite result (Figs. 5Y–5BB). In summary, KIF4A could affect the apoptosis process of MG63 and U20S cells.

Figure 5 Silencing KIF4A induced apoptosis and cell cycle arrest of Osteosarcoma cells.

(A–L) Annexin V-FITC and PI was used to evaluate apoptosis rate. (M–X) Cell cycle was detected by flow cytometry. (Y–BB)The expression of apoptosis-related proteins Bax and Bcl-2 and caspase-3 were determined by Western blot analysis. *P < 0.05 compared with Control, #P < 0.05 compared with si-NC, &P < 0.05 compared with oe-NC.

Discussion

Kinesin superfamily(KIF) proteins play a key role in cellular functions, such as mitosis and intracellular transport of organelles and vesicles, and there are 14 recognized kinesin families (kinesin 1–14A/B) (Jungwirth et al., 2019). Upregulated expression of KIF proteins leads to premature separation of chromosomes, which can cause progression of cancer(Manning, Hooper & Sahai, 2015; Kato et al., 2016). The KIF4A gene encodes the chromokinesin protein KIF4A, an ATP-dependent molecular motor that promotes mitotic chromosome condensation and segregation (Mazumdar, Sundareshan & Misteli, 2004). KIF4A can also directly bind to chromatin and participate in DNA damage repair by associating with BRCA2 (Wu & Chen, 2008; Wu et al., 2008). Many studies have shown that the expression of KIF4A in liver cancer, cervical cancer and oral cancer tissues is up-regulated, and is positively correlated with poor prognosis (Hou et al., 2017; Narayan et al., 2007; Taniwaki et al., 2007; Minakawa et al., 2013). KIF4A overexpression might promote lung cancer resistance to cisplatin, while in breast cancer cells, its overexpression promotes cell apoptosis during treatment with doxorubicin (Wang et al., 2014).

Our bioinformatics analysis results showed that overexpression of KIF4A played a role in mitosis and could become a potential new diagnostic and prognostic marker for various cancers. The observed pan-cancer upregulation of KIF4A is probably an indicator of the increased mitotic rate in cancer cells. High KIF4A expression was correlated with poor overall survival of OS patients. In vitro experiments showed that silencing KIF4A could inhibit OS cell viability, colony formation, migration and invasion, and induce apoptosis and cell cycle arrest.

β-Catenin, the core mediator of Wnt/β-catenin signaling pathway, is regulated by GSK-3β-promoted phosphorylation and ubiquitin/proteasome pathway degradation. β-catenin mutation could lead to the abnormal activation of Wnt target gene, leading to the occurrence of cancer (Qi & Yang, 2020). The Wnt/β-catenin signaling pathway plays an important role in chromosomal instability (Hadjihannas & Behrens, 2006). Various studies have proven that the Wnt/β-catenin signaling pathway is implicated in the pathogenesis of cancers, especially OS (Chen, Zhao & Fan, 2020; Li et al., 2020; Yin et al., 2020; Zhao et al., 2020). Our results indicated that KIF4A modulates the Wnt/β-catenin pathway in OS cells.

In conclusion, KIF4A could be used as a biomarker and prognostic indicator for pan-cancer diagnostic. KIF4A could regulate Wnt/β-catenin signaling as a new target for OS therapy.

Supplemental Information

Supplemental Information 1 Expression profile of KIF4A in cancer tissue and paired normal tissue

Click here for additional data file.

Supplemental Information 2 Functional enrichment of the 28 pan-cancer DEGs via STRING analysis

Click here for additional data file.

Supplemental Information 3 Raw data: KIF4A mRNA level was higher in OS cells than in the normal osteoblast hFOB1.19

Click here for additional data file.

Supplemental Information 4 Raw data: KIF4A protein expression was significantly higher in OS cell lines than in the normal osteoblast hFOB1.19

Click here for additional data file.

Supplemental Information 5 Expression of KIF4A in cells was detected by RT-PCR

Click here for additional data file.

Supplemental Information 6 Expression of KIF4A in cells was detected by WB

Click here for additional data file.

Supplemental Information 7 CCK8 was used to detect cell proliferation

Click here for additional data file.

Supplemental Information 8 Clone formation assay was used to detect the OD value of cloned cells

Click here for additional data file.

Supplemental Information 9 Transwell assay was used to test the invasion

Click here for additional data file.

Supplemental Information 10 Wound healing assay was utilized to detect the migration

Click here for additional data file.

Supplemental Information 11 Annexin V-FITC and PI was used to evaluate apoptosis rate

Click here for additional data file.

Supplemental Information 12 Cell cycle of MG63 was detected by flow cytometry

Click here for additional data file.

Supplemental Information 13 Cell cycle of U20S was detected by flow cytometry

Click here for additional data file.

Supplemental Information 14 The expression of apoptosis-related proteins Bax and Bcl-2 and caspase-3 were determined by Western blot analysis

Click here for additional data file.

Additional Information and Declarations

Competing Interests

Author Contributions

Data Availability

The authors declare there are no competing interests.

Jiankang Pan conceived and designed the experiments, performed the experiments, analyzed the data, prepared figures and/or tables, and approved the final draft.

Xiaohua Lei conceived and designed the experiments, performed the experiments, analyzed the data, prepared figures and/or tables, authored or reviewed drafts of the paper, and approved the final draft.

Xinzhan Mao conceived and designed the experiments, authored or reviewed drafts of the paper, and approved the final draft.

The following information was supplied regarding data availability:

The raw measurements are available in the Supplementary Files.

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
