# Peer review of "Identification of KIF4A as a pan-cancer diagnostic and prognostic biomarker via bioinformatics analysis and validation in osteosarcoma cell lines"

_PeerJ, doi:10.7717/peerj.11455_

## Round 0.1 · original submission · Major Revisions

Dear authors,

As you will see your work has now being reviewed by 3 Reviewers. All of them say that the work needs strong improvements and also additional experiments are suggested. Clarification of the objectives is also necessary. The authors should also compare the predictive performance of their study with previous works on the same problem and databases.

Please address All the Reviewers' concerns in a point-by-point letter.
Best regards

Reviewer 1 ·

Basic reporting

no comment

Experimental design

Line 128: Please mention how the crystal violet dye was dissolved before measuring the absorbance

Validity of the findings

Line 196: Fig 5C does indicate statistical significance. Please run stats on multiple blots ran on different days and provide statistics. This also applies to Fig 6 (line 202)

Additional comments

Line (220-222): Please silence KIF4A in the non-cancer cell line and compare the effects with the current OS cell-lines.
Also, it is important to exogenously overexpress KIF4A of a constitutively active promoter in the KIF4A silenced OS cell-lines to see if it rescues inhibition of cell viability, migration and reverses apoptosis. This will significantly improve the quality of the manuscript.

Reviewer 2 ·

Basic reporting

Overall, the manuscript is concise and clear. The authors should carefully describe the results in the Results section with sufficient details such as p-value.

Experimental design

The authors addressed the research questions by appropriate experimental design. Methods had been described with details. It can be improved by providing more specific information of some materials used in the experiment. Please find the details in the comments.

Validity of the findings

The scientific findings are sound.

Additional comments

KIF4A has been implicated in multiple types of cancer, such as prostate cancer, lung cancer, colorectal carcinoma, etc. To investigate how KIF4A affects cancer cell progression, Pan and colleagues compared gene expression from 12 cancer types. The authors identified KIF4A significantly increased in 11 different types of cancer. The authors confirmed the upregulation of KIF4A in different types of osteosarcoma cells. Silencing KIF4A via siRNA significantly decreased cell proliferation and increased cell apoptosis. Overall, the manuscript is concise. The study can be improved with the following specific points being appropriately addressed.
1. What is the efficiency of siRNA transfection? Besides the qPCR, did the authors confirm the downregulation of KIF4A by Western blot?
2. Did the siRNA of the KIF4A downregulate the expression of KIF4A to the same level as the normal osteoblast hFOB1.19 cells?
3. Did you compare the cell proliferation and apoptosis of KIF4A knockdown with the normal osteoblast cells?
4. Please specify the Wnt antibody. Is it Wnt3a?
5. Did you test the abundance of phosphorylated Beta-catenin?
6. In the figure 5, the bar graph did not math with the FACS figures.

Reviewer 3 ·

Basic reporting

The authors used bioinformatics to find the KIF4A as a pan-cancer diagnostic and prognostic biomarker. The osteosarcoma cell lines have been retrieved to validate the performance results. The idea is of interest, but there are major points that need to be addressed to meet the quality requirement of the journal.

- English should be improved.

- There lacks of literature review on bioinformatics-based OS analysis.

- Abstract is written unclearly. It did not emphasize the hypotheses and objective of this study.

- From lines 53-57, I suggest not to include them in the "Introduction" since it mentioned the results.

- Abbreviation should be defined at the first appearance.

Experimental design

- Source codes should be provided for replicating the methods.

- Did the authors concern about the batch effect removal in their datasets?

- TCGA has been used in previous bioinformatics studies such as PMID: 32942564 and PMID: 33086550. Thus, it is suggested to refer to more works in this description.

Validity of the findings

- The authors should compare the predictive performance with the previous works on the same problem/data.

- Survival analysis has been conducted. Why did the authors not use Hazard Cox analysis?

- The authors should have additional validation data related to RNA-seq or gene expression level.

Additional comments

No comment.

---

## Round 0.2 · accepted · Accept

Dear authors,

Thank you for the modifications done to your manuscript.

All the reviewers consider that it is now acceptable for publication.

Congratulations.

Reviewer 1 ·

Basic reporting

no comment

Experimental design

no comment

Validity of the findings

no comment

Additional comments

Thanks for addressing all the comments or concerns. The manuscript looks fine.

Reviewer 2 ·

Basic reporting

The authors have addressed my concerns.

Experimental design

The authors have addressed my concerns.

Validity of the findings

The authors have addressed my concerns.

Additional comments

The authors have addressed my concerns.

Reviewer 3 ·

Basic reporting

No comment.

Experimental design

No comment.

Validity of the findings

No comment.

Additional comments

My previous comments have been addressed well.